# Research on Audit Supervision of Internet Finance

**Hua Liu [1] and Sheng Ge [2],***

[1]    School of Finance, Nanjing Audit University, Nanjing 211815, China; huazi1029@sohu.com
[2]    School of International Education, Nanjing Audit University, Nanjing 211815, China
*    Correspondence: gesheng@139.com

**Abstract:** Internet finance is a new form of finance that applies capacities found on the Internet to the traditional financial industry. However, at the present stage, internet finance is faced with many problems, such as overly rapid development and non-standard operation. This paper adopted the evolutionary game theory as the analysis tool to design an evolutionary game model of government audit supervision of Internet finance, and analyzed the evolutionary stability of the strategies used by Internet financial institutions and government financial audit supervision departments. A simulation calculation was carried out by placing the calculation experimental method "Scenario–Coping", which simulated the initial probability of different strategies adopted by both parties of the game and evaluated the influence of changing the penalty intensity of Internet financial institutions' violation on the outcome of the evolutionary game. Based on the simulation analysis, the paper provided policy suggestions on strengthening audit supervision and promoting its sustainable development from three aspects: strengthening the construction of the Internet financial credit information system, improving Internet financial laws and regulations, and improving the early warning level of Internet financial credit risk.

**Keywords:** internet finance; government auditing; evolutionary game; system simulation; computing experiment

## 1. Introduction

In recent years, driven by the needs of financial markets, the Internet has developed rapid systems and applications useful to the financial industry. As a result, internet finance has sprung up like bamboo shoots after a rain, and has become an emerging force in China's financial market. In 2013, the first year of the Internet finance era, Internet thinking swept through traditional forms and structures of finance like the Renaissance. Most traditional financial institutions including banks, securities and fund companies, and insurance companies started to make significant arrangements to retain their competitive edge, while the e-commerce giants such as Alibaba, Tencent, Baidu and Sina began to create their own business models hoping to establish their own Internet empire. With the traditional business models and financial structure being challenged, a financial "New Normal" in the context of "Internet Plus" was being formed.

Although all levels of the Chinese government encourage innovative development of Internet finance, great risks still lie ahead since both the Internet and finance are industries with high risks. A combination of both industries poses even higher risks. Internet finance is hence faced with a wide range of risks including systematic risk, liquidity risk, credit risk, technological risk as well as operational risk. Thus, strong government audit supervision is needed to ensure the healthy development of Internet finance.

Internet finance is supported by Internet technology and is governed by the market. Higher risks seem inevitable. The rapid development of Internet finance brings about moral hazards and adverse

selection in information safety. At present, the Internet financial credit information system in China is yet to be perfected and the legal restraint on Internet finance is largely flawed. As a result, problems such as Internet fraud and credit default arise (Hong and Cao 2014). In terms of Internet financial credit risk, Duarte et al. (2010) analyzed the role credit plays in financial transactions and proved that even in countries with sound legal systems, the problem of credit was still unavoidable (Duarte et al. 2010). Agarwal and Hauswald (2008) contended that online financing platforms played a positive role in helping small and micro-sized enterprises solve financing problems, because small enterprises were unable to obtain loans from commercial banks due to their failure to provide public credit ratings, and they could only turn to the online loan market for help (Agarwal and Hauswald 2008). By virtue of its unique role, the peer-to-peer (P2P) network loan has emerged on a large scale in the world and become a supplement to traditional finance. Economides (2001) pointed out that this new form of borrowing reduced transaction costs and greatly expanded the coverage of financial services (Economides 2001). Empirical research on P2P online lending shows that, in terms of the characteristics of borrowers, the main objects of P2P loans are short-term and small borrowers who are rejected by traditional financial institutions because of the inaccuracy of credit information or the failure to provide sufficient collateral, and most of them are from the working class with poor credit (Zhuang and Zhou 2015). Based on order statistics released "on credit", so far the average loan amount is about 7371.795 Yuan, borrowing on average to attract investors to participate in the tender number is 19.9148, average borrowing success rate is 0.3252, the duration of the loan is an average of 6.5 months, and most of the P2P borrowers' loan time limit is in the following 1 year (Guo 2012; Wang and Liao 2014).

The top priority in maintaining financial stability is to prevent financial risks. However, the majority of domestic research focuses on introducing various new models of Internet financial loan, whereas financial auditing in China mainly focuses on investigating cases that have violated laws and rules. There have been few in-depth discussions on the early warning of financial risks. At present, the priority of financial auditing is to improve laws and regulations, guarantee the quality of assets, conduct effective internal control, and improve the authenticity of accounting information (Liu 2002, 2013). In current research on the early warning of financial risks, the Probit Model, the Sachs-Tornell-Velasco (STV) cross-sectional regression model, and the Kaminsky–Lizondo–Reinhart (KLR) signal approach have been widely accepted (Zhang and Sun 2003). With the development of financial auditing, new approaches such as VaR and Pressure Test have been adopted (Xu and Xu 2010). However, a comprehensive research on the early warning of Internet financial risk is still lacking. In 2017, Liu Hua and Zhang Jie took the P2P online lending platforms as an example and established the P2P online lending credit risk evaluation index system within the framework of their "auditing immune system" analysis. They conducted an empirical study on 10 P2P online lending platforms by combining the entropy weight method and grey correlation integrated evaluation model, and proposed an auditing and governance strategy to prevent Internet financial risk (Liu and Zhang 2017).

Computational experiment finance (CEF) is a simulation method that aims to reveal the macro operational law of the financial market by (1) setting up the specific learning modes and behavioral mechanisms of microscopic financial bodies, and (2) including the communication and interaction among microscopic financial bodies under a given financial system and financial market with the aid of modern computer science and technology (Lebaron 2000; Levy et al. 2000). Arthur et al. (2007) pioneered computing experiment finance (Arthur et al. 2007). They conducted research on the stock market by substituting a computer simulation model based on Agent for the original mathematical analytic model, and established the so-called Artificial Stock Market (ASM). Their work marked the birth of a new branch of finance—Computing Experiment Finance Based on Agent. Zhang et al. (2003) was the first Chinese scholar who systematically introduced the theoretical foundation, basic concepts and research areas of Computing Experiment Finance. Zhang explained the modeling approach by using the Santa Fe Institute's ASM as an example, and explored its relationship with other branches such as the Financial Market Microstructure Theory in Modern Finance and Behavioral Finance

(Zhang et al. 2003). Many scholars have since applied the approach of the computing experiment to the field of financial risk supervision. Pérez-Martín and Vaca (2017) calculated the credit risk of banks in housing mortgage and the mortgage default rates by using computing experiment, and compared the computational efficiency of different statistical and data-mining methods (Pérez-Martín and Vaca 2017). Liu and Chen (2018) adopted evolutionary game theory as the analysis tool to design the evolutionary game model of collaborative governance and auditing of local governments' debts, and analyzed the evolutionary stability of the strategies of the audit offices and local governments (Liu and Chen 2018). On this basis, the simulation calculation was carried out by using a calculation experimental method "Scenario–Coping", which simulated the initial probability of different strategies adopted by both parties of the game, and evaluated the influence of the soundness of three sub-mechanisms on the outcome of the evolutionary game.

In sum, study abroad of P2P lending not only carried out thorough research in theory (Mr Stiglitz and Weiss 2016), but also from the perspective of empirical analysis of the data model calculation and practical test detection (Yum et al. 2015), the risk of overall lending to P2P networks, and the success rate of the borrower and the lender investment decision problem has received relatively effective research and demonstration (Stiglitz and Weiss 2016; Yum et al. 2015). However, the majority of research on Internet finance focuses on specific problems in actual business operation. Research on the early warning of Internet financial credit risk has started relatively late and focused mostly on interpreting of the concept of Internet financial risk, and summarizing and evaluating western early warning theories. Although many Chinese scholars have optimized the early warning model according to specific national conditions, there a systematic and quantitative analysis is still lacking due to insufficient statistics and materials available regarding the supervision of Internet financial risk. In such a context, we adopted the computing experiment approach, and analyzed the evolutionary game of the government audit supervision of Internet finance. Based on the calculation experimental method "Scenario–Coping", we extracted correlative factors of both parties of the game, and simulated the initial probability of different strategies adopted by both parties of the game, and evaluated the influence of changing the penalty intensity of Internet financial institutions' violation on the outcome of the evolutionary game. Our results are meaningful for holding the bottom line of preventing systematic financial risk and improving the healthy development of China's Internet finance.

## 2. Materials and Methods

### 2.1. Behavioral Analysis of the Government Audit Supervision of Internet Finance

In the context of "Internet Plus", Internet mercantile services rapidly integrate with finance, and Internet finance operation modes overlap with traditional finance in many fields. Although it is a combination of finance and Internet technology, Internet finance is still finance by nature and follows the functions of finance. Thus, it runs with not only the typical risks in traditional finance, but also its own unique risks. Banks and other financial institutions pay special attention to systematic risk, liquidity risk, and credit risk. Internet finance institutions should do the same, especially for credit risk, since Internet financial transactions are still based on credit risk pricing. Problems such as information asymmetry and information leakage in traditional finance are still ubiquitous in Internet finance. These risks can be increasing since Internet finance is still in an early stage in China and the related laws and regulations are a work in progress. For example, a P2P online lending platform can face a high credit risk. A typical kind of fraud is the "Ponzi Scheme", which, in simple terms, is a fraudulent investing scam promising high rates of return with little risk to investors. The Ponzi scheme generates returns for earlier investors by acquiring new investors through creating a false impression that the company actually makes profits. Once there are no new investors, or the funds contributed by the new investors are not sufficient to pay the "interest" of the first investors, the "Ponzi Scheme" will not be able to continue.

With the rapid development of Internet finance, many problems arise, such as information asymmetry, lack of information transparency, lack of supervision, and information technology risk. In order to prevent these risks, the China Banking Regulatory Commission has established the principle of "innovative supervision, moderate supervision, classified supervision, collaborative supervision" and "the combination of government supervision and self-regulation". Even so, the current supervision of Internet finance is inadequate. Some unscrupulous institutions play games with the legal system and exploit the holes in regulations. Some risk-control institutions conduct misleading promotion and illegal financing under the disguise of Internet finance. As a result, Internet finance supervision requires an authoritative, independent, and unified supervision system. Such requirements can only be met by Internet finance auditing.

Liu Jiayi, the former Auditor General of the China National Audit Office, once indicated to the National Auditing Conference that "modern government audit is the immune system for the functioning of the economy and society", from which the theory of the government auditing immune system is derived. The establishment of Internet finance audit supervision system could help avoid the aimlessness in government auditing processes, ensure reasonable allocation of audit resources, and improve the efficiency of auditing to a large extent. In this paper, we aimed to provide a new method for the establishment of an Internet finance audit supervision system by combining the theoretical framework of audit "immune system" and of Internet finance supervision (Figure 1).

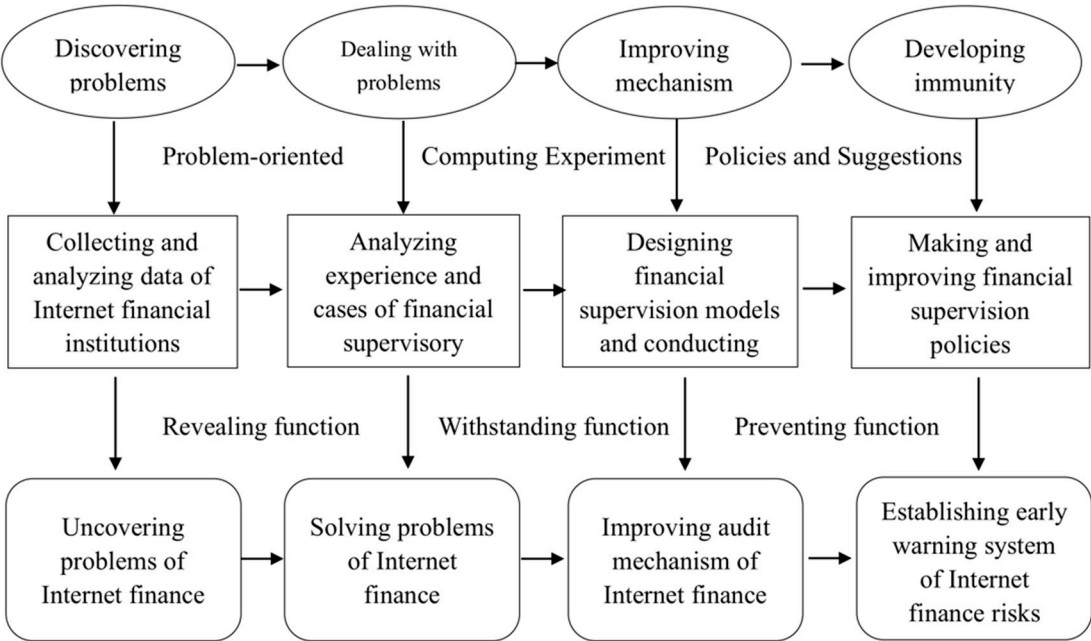

**Figure 1.** "Immune System" analysis framework of Internet finance government audit supervision.

As Figure 1 shows, the way government auditing acts as an "immune system" is to enhance immunity by discovering problems, solving problems and perfecting mechanisms, with each step closely linked with and mutually dependent on the others. We applied the theory of "immune system" to the process of Internet finance audit supervision. Adopting a problem-oriented approach, we: (1) discovered the problems and risks of Internet finance due to excessive growth and lack of supervision; (2) analyzed the relationship between Internet financial institutions and financial audit offices as two parties of the game as well as their gains and losses through the approach of computing experiment and case study of financial supervisory authorities; (3) revealed different strategies chosen by both parties under different circumstances in order to bring the defense capability of the "immune system" into full play; and, (4) suggested a more targeted policy mix for Internet finance audit supervision so as to give early warning to Internet financial risks.

### 2.2. Evolutionary Game Analysis of Internet Finance Government Audit Supervision

A behavioral analysis of Internet finance government audit supervision shows that there is information asymmetry between Internet financial institutions and financial audit offices. In accordance with the various audit strategies adopted by audit offices, Internet financial institutions choose to, or not to, comply with laws and regulations based on their own situations. The behaviors of both entities form a bounded rational dynamic game. For the convenience of adopting a game model for analysis, and on the basis of behavioral characteristics of Internet financial institutions and financial audit offices as well as the status quo of Internet finance government audit supervision, we propose the following hypotheses:

1.  As an endogenous system of national governance, government auditing plays a crucial role in preventing Internet financial risks and safeguarding national financial security (Cao and Xiong 2018). As one party of the game, "financial audit offices" are national audit departments which carry out auditing and supervision on the other party to the game (Internet financial institutions) on behalf of the public and the central government. The decisions made by government financial audit are in real time, rather than on a regular basis, which means auditors are able to make decisions at any time based on the situations of Internet financial institutions, and carry out audit immediately. This hypothesis guarantees the consistency of the game process.
2.  As another party of the game, "Internet financial institutions" are companies involved in providing financial services using Internet technology. The healthy development of Internet finance needs supervision from government audit. Their legitimate business operations help to boost local economic development and enhance the effectiveness of government audit offices (Zhou 2017). If compliance is not emphasized, the subjects of liability will be held accountable and penalized after audit offices conduct audits and discover problems. These financial institutions subsequently will be unable to make profits from business operations and might receive harsh penalties.

Based on the above hypotheses, game utility matrix for both parties of the game under different strategy choices is shown in Table 1 and the variables therein are set as follows (all the variables used in the evolutionary game matrix are indexed in Appendix A at the end of the present paper):

1.  Suppose the profits made by Internet financial institutions through legitimate business operations are $I_1$, their expenditure is $C_1$, bonuses, the rewards gained through legitimate business operations are $D_1$, and fines imposed by audit offices due to non-compliance are $L$.
2.  Suppose the expenditure for carrying out audit supervision by the financial audit supervisory departments is $C_2$, and the social economic benefits resulting from the cooperation in audit supervision between Internet financial institutions and financial audit supervisory departments are $I_2$. Due to the scarcity of auditing resources, it is impossible for government audit offices to focus solely on Internet financial institutions. The compliance of the latter helps to create a positive environment for financial operations, which in turn reduces the pressure on financial audit departments and generates an opportunistic benefit ($D_2$).
3.  Suppose the probability of choosing "compliance" and "noncompliance" by Internet financial institutions in the game is x (where $0 < x < 1$) and $1 - x$, respectively; the probability of carrying out "audit supervision" and not performing this function by the financial audit offices is y (where $0 < y < 1$) and 1-*y* (respectively).

We established the evolutionary game model using the Game Utility Matrix of Internet Finance Government Audit Supervision:

**Table 1.** Game utility matrix of Internet finance government audit supervision.

| Strategies Chosen by Both Parties of the Game | | Financial Audit Supervisory Departments | |
|---|---|---|---|
| | | **Carrying out Audit Supervision y** | **not Carrying out Audit Supervision $1 - y$** |
| Internet financial institutions | compliance $x$ | $I_1 - C_2 + D_1, I_2 - C_2$ | $I_1 - C_1, D_2$ |
| | noncompliance $1 - x$ | $-L, -C_2$ | $0, 0$ |

### 2.2.1. Utility Model of Internet Financial Institutions

When Internet financial institutions choose the strategy of "compliance", the utility is set as $u11$; when they choose the strategy of "non-compliance", the utility is set as $u12$. Thus, the following equations can be proposed:

$$u_{11} = y(I_1 - C_1 + D_1) + (1 - y)(I_1 - C_1) \tag{1}$$

$$u_{12} = -yL \tag{2}$$

If Internet financial institutions combine both strategies, the average expected utility of both strategies is set as $u_1$. The following equation can be proposed:

$$u_1 = xu_{11} + (1 - x)u_{12} \tag{3}$$

### 2.2.2. Utility Model of Financial Audit Supervisory Departments

When financial audit supervisory departments choose to carry out audit supervision, the utility is set as $u_{21}$; when they choose not to carry out audit supervision, the utility is set as $u_{22}$. The following equations can be proposed:

$$u_{21} = x(I_2 - C_2) + (1 - x)(-C_2) \tag{4}$$

$$u_{22} = xD_2 \tag{5}$$

If financial audit supervisory departments combine both strategies, the average expected utility of both strategies is set as $u_2$. The following equation can be proposed:

$$u_2 = yu_{21} + (1 - y)u_{22} \tag{6}$$

## 3. Results

According to the Malthusian dynamic equation, the dynamic change rate of the probability of Internet financial institutions choosing the strategy of "compliance" is $\frac{dx}{dt}$, the dynamic change rate of the probability of financial audit supervisory departments choosing to carry out audit supervision is $\frac{dy}{dt}$, and the following replicator dynamics equations can be proposed:

$$\frac{dx}{dt} = x(u_{11} - u_1) = x(1 - x)(yD_1 + I_1 - C_1 + yL) \tag{7}$$

$$\frac{dy}{dt} = y(u_{21} - u_2) = y(1 - y)(xI_2 - C_2 - xD_2) \tag{8}$$

Equations (7) and (8) show the rate and direction of game learning by Internet financial institutions and financial audit supervisory departments. When both replicator dynamics equations equal zero, the Internet finance government audit supervision game achieves a state of relative stability and balance.

If $F(x) = \frac{dx}{dt} = 0$, $F(y) = \frac{dy}{dt} = 0$, then $x = \frac{C_2}{I_2 - D_2}$, $y = \frac{C_1 - I_1}{D_1 + L}$ ($0 \leq \frac{C_2}{I_2 - D_2} \leq 1$, $0 \leq \frac{C_1 - I_1}{D_1 + L} \leq 1$). According to the utility function of Internet financial institutions and financial audit supervisory departments, the variation trends of $x$ and $y$ can be described as follows:

*3.1. Analysis of Internet Financial Institutions Stability*

1.  When $y = \frac{C_1 - I_1}{D_1 + L}$, $F(x) \equiv 0$. $X$ is the evolutionary stability strategy of Internet financial institutions.
2.  When $y \neq \frac{C_1 - I_1}{D_1 + L}$, based on the stability theorem of differential equations and the nature of evolutionary stability strategy, it can be inferred that $x^*$ is the evolutionary stability strategy if $F(x^*)' < 0$.

   Thus, when $y > \frac{C_1 - I_1}{D_1 + L}$, $F(x)'_{x=0} > 0$, $F(x)'_{x=1} < 0$, $x = 1$ is the evolutionary stability strategy of Internet financial institutions; when $y < \frac{C_1 - I_1}{D_1 + L}$, $F(x)'_{x=0} < 0$, $F(x)'_{x=1} > 0$, $x = 0$ is the evolutionary stability strategy of Internet financial institutions.

*3.2. Analysis of Financial Audit Supervisory Departments' Stability*

1.  When $x = \frac{C_2}{I_2 - D_2}$, $F(y) \equiv 0$. $Y$ is the evolutionary stability strategy of financial audit supervisory departments.
2.  When $x \neq \frac{C_2}{I_2 - D_2}$, based on the stability theorem of differential equation and the nature of evolutionary stability strategy, it can be inferred that $y^*$ is the evolutionary stability strategy if $F(y^*)' < 0$.

   Thus, when $x > \frac{C_2}{I_2 - D_2}$, $F(y)'_{y=0} > 0$, $F(y)'_{y=1} < 0$, $y = 1$ is the evolutionary stability strategy of financial audit supervisory departments; when $x < \frac{C_2}{I_2 - D_2}$, $F(y)'_{y=0} < 0$, $F(y)'_{y=1} > 0$, $y = 0$ is the evolutionary stability strategy of financial audit supervisory departments.

*3.3. Evolutionary Stability Analysis of Both Parties of the Game*

   Based on the above strategy, evolutionary stability analysis and the replicator dynamics equations of both parties, the following graph can be drawn.

   From Figure 2, it can be determined that there are five equilibrium points, that is, A(0,1), B(1,1), C(1,0), D(0,0) and E($\frac{C_2}{I_2 - D_2}$, $\frac{C_1 - I_1}{D_1 + L}$). Among them, A(0,1) and C(1,0) are unstable equilibrium points, E($\frac{C_2}{I_2 - D_2}$, $\frac{C_1 - I_1}{D_1 + L}$) is the saddle point, and B(1,1) and D(0,0) are evolutionarily stable equilibrium points. An analysis of the five equilibrium points shows that when financial audit supervisory departments carry out audit and supervision extensively, Internet financial institutions will choose "compliance" as the optimal strategy after a certain period of time. Meanwhile, if Internet financial institutions choose compliance, it is certain that financial audit supervisory departments are willing to offer audit recommendations so as to enhance risk management of Internet finance and help them to gain more benefits from better governance.

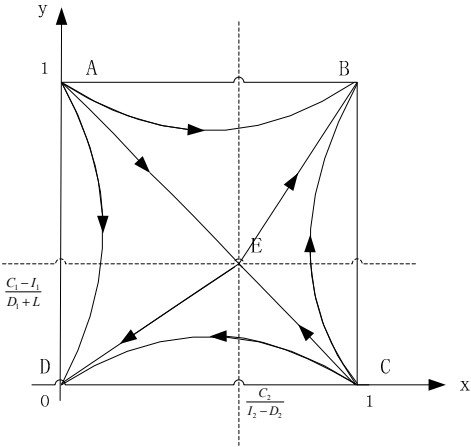

**Figure 2.** Evolutionary phase diagram of Internet finance government audit supervision game.

*3.4. Evolutionary Analysis of Model Parameters*

The above analysis shows that the evolution of the gaming system can be driven towards the expected direction by changing the parameters of both parties under different strategy choices, followed by the change of $\frac{C_2}{I_2-D_2}$ and $\frac{C_1-I_1}{D_1+L}$, and the areas of Quadrangle ADCE and Quadrangle ABCE. The following explanations of the model are developed by taking cost, benefit and penalty as examples.

1.  The parameter of cost. $C_1$ and $C_2$ represent the cost of choosing "compliance" by Internet financial institutions and choosing to carry out audit supervision by financial audit supervisory departments. As $C_1$ decreases, $\frac{C_1-I_1}{D_1+L}$ decreases and point E moves downward. The replicator dynamics phase diagram shows that, under such a condition, the area of Quadrangle ADCE decreases while the area of Quadrangle ABCE increases, which means there is a higher probability that the initial state is in Quadrangle ABCE and a higher probability that the gaming system will evolve to the equilibrium strategy (1,1). Similarly, as $C_2$ decreases, $\frac{C_2}{I_2-D_2}$ decreases and point E moves towards the left, resulting in a higher probability that the gaming system will evolve to the equilibrium strategy (1,1). The above analysis shows that in the game of Internet finance government audit supervision, the lower the costs of both parties are, the higher the probability of the system converging to (1,1).

2.  The parameters of benefit. $I_1$ and $I_2$ represent the benefit of choosing "compliance" by Internet financial institutions and choosing to carry out audit supervision by financial audit supervisory departments. As $I_1$ increases, $\frac{C_1-I_1}{D_1+L}$ decreases and point E moves downward. There is a higher probability that the gaming system will evolve to the equilibrium strategy (1,1). Similarly, as $I_2$ increases, $\frac{C_2}{I_2-D_2}$ decreases and point E moves towards the left, leading to a higher probability that the gaming system will evolve to the equilibrium strategy (1,1). The above analysis shows that in the game of Internet finance government audit supervision, the higher the benefits of both parties are, the higher is the probability of the system converging to (1,1).

3.  The parameter of penalty. L represents the penalty imposed on Internet financial institutions when they choose "noncompliance" and financial audit supervisory departments choose to carry out audit supervision. As $L$ increases, $\frac{C_1-I_1}{D_1+L}$ decreases, $\frac{C_2}{I_2-D_2}$ remains the same and point E moves downward, resulting in a higher probability that the gaming system will evolve to the equilibrium strategy (1,1). When financial audit supervisory departments carry out audit supervision and discover non-compliance of Internet financial institutions, the latter will face heavier losses and in turn they will have a stronger motivation to choose compliance so as to reduce losses. As a result, the system converges to (1,1).

## 4. Discussion

To conduct in-depth research on the strategy evolution of both parties of the game and the influence relevant parameters have on the strategies of both parties, the College of Audit Officials of China National Audit Office and Nanjing Audit University collaborated to develop a proposal for Internet finance government audit supervision in June 2018. Field research had been conducted on the current situations of Internet finance government audit supervision in Jiangsu province. The initial value of the game model is set as the preset original data. The financial audit office in this region plans to conduct a special audit on Internet finance and the cost per month is 40,000 RMB. The social economic benefit resulting from the cooperation in audit supervision between Internet financial institutions and financial audit supervisory departments is 40,000 RMB per month. The compliance of the Internet financial institutions helps to create a positive environment for financial operations, which in turn reduces the pressure on financial audit departments and generates an opportunity benefit equal to 20,000 RMB per month. Field research has shown that most Internet financial institutions in this region expand their market and attract new clients by sacrificing profits to increase market share. Suppose the average monthly cost of the Internet financial institutions in this region for legitimate business operation is 50,000 RMB, and the profit they make from legitimate business operations is 40,000 RMB

per month. Due to the government's encouragement of innovation and the development of Internet finance, an extra benefit of 30,000 RMB per month can be made by an Internet financial institution from the government and market. If noncompliance is discovered by the financial audit supervisory department, 20,000 RMB per month will be fined.

The initial parameters of the model are set as follows: $I_1 = 4$, $C_1 = 5$, $D_1 = 3$, $D_2 = 2$, $I_2 = 10$, $C_2 = 4$, $L = 2$. Based on computing experiment simulation platform and the analysis of evolutionary game model (INITIAL TIME = 0, FINAL TIME = 100, TIME STEP = 1), we conducted the formulation calculation as follows:

1.  The influence of the initial value of x (the probability of Internet financial institutions strategy choice). According to our analysis of the evolutionary game model, in this example, $x^* = \frac{C_2}{I_2-D_2} = 0.5$ and $y^* = \frac{C_1-I_1}{D_1+L} = 0.2$. When $0.2 < y < 1$, if the initial value of x is higher than 0.5, then $x = 1$ is the evolutionary stable point; otherwise, $x = 0$ is the evolutionary stable point. Suppose the initial value of y is 0.3 and the variation of the initial value of $x$ is between 0.1 and 0.9. If one simulation is carried out whenever the initial value of $x$ changes by 0.1, the simulation results are as shown in Figures 3 and 4.

2.  The influence of the initial value of $y$ (the probability of financial audit supervisory departments' strategy choice). According to our analysis of the evolutionary game model, in this example, when $0.5 < x < 1$, if the initial value of y is higher than 0.2, then $y = 1$ is the evolutionary stable point; otherwise, $y = 0$ is the evolutionary stable point. Suppose the initial value of x is 0.6 and the variation of the initial value of x is between 0.1 and 0.9. If one simulation is carried out whenever the initial value of $y$ changes by 0.1, the simulation results are as shown in Figures 5 and 6.

The analysis of the above two computing experiment examples shows that in the process of evolutionary game, the strategy choice of both parties of the game affect each other. Only when $x > \frac{C_2}{I_2-D_2}$ and $y > \frac{C_1-I_1}{D_1+L}$ does the game simulation system converge to (1,1). Specifically speaking, when $y > \frac{C_1-I_1}{D_1+L}$, x converges to 1; otherwise, x converges to 0. When $x > \frac{C_2}{I_2-D_2}$, y converges to 1; otherwise, y converges to 0. These results suggest that as long as financial audit supervisory departments keep carrying out audit supervision on Internet financial institutions with a probability higher than $\frac{C_1-I_1}{D_1+L}$, the game result will converge to the result of compliance and audit supervision. The Internet financial risks can, thus, be effectively managed and prevented, and the health and sustainable development of Internet finance can be ensured.

3.  The influence of penalties on Internet financial institutions. To simulate the influence of the severity of punishment of Internet financial institutions by financial audit supervisory departments, suppose the initial value of $x$ is 0.6, the initial value of $y$ is 0.3, and the other variables remain constant but the amount of fines increases gradually from $L = 1$. We conducted three simulations ($L = 4$, 7, and 10); the results appear in Figures 7 and 8.

A comparison of the above simulation results and the different plotted curves of $L$ shows that when $x > \frac{C_2}{I_2-D_2}$ and $y > \frac{C_1-I_1}{D_1+L}$, as the penalty L gradually increases, the evolution of $x$ and $y$ converges to 1. The simulation results suggest that the decisions made by both Internet financial institutions and financial audit supervisory departments are influenced by the severity of punishment. The larger the amount of the fine, the faster the game result between Internet financial institutions and financial audit supervisory departments converges to compliance and audit supervision, respectively.

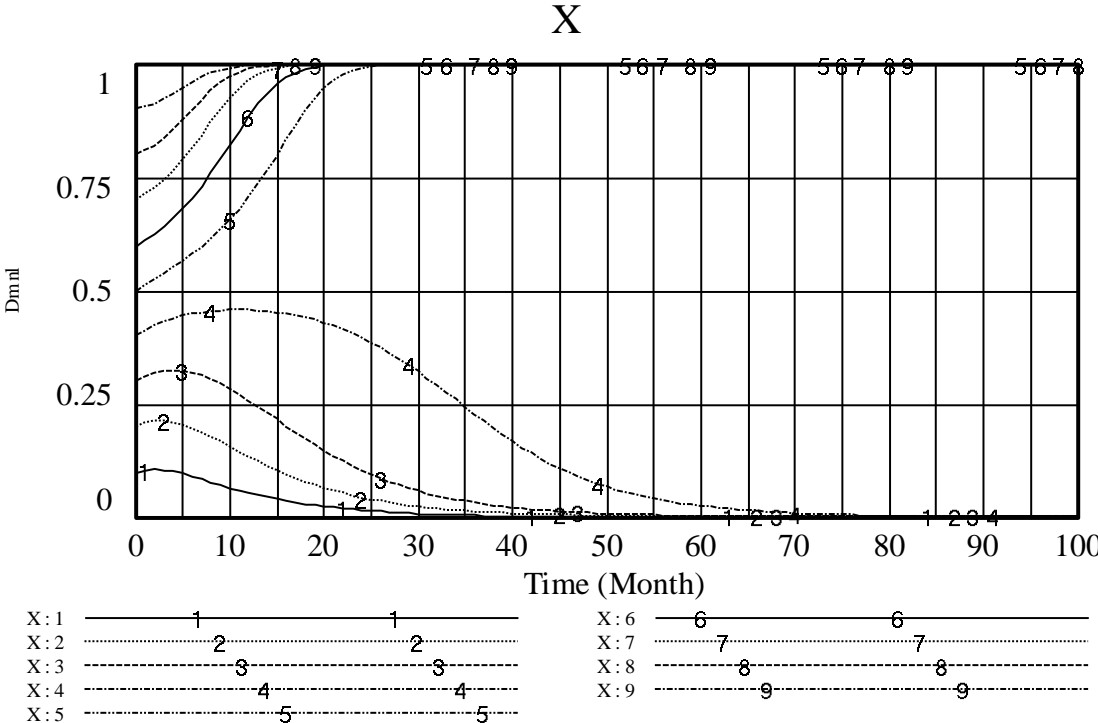

**Figure 3.** Simulation result of x when initial value of y is 0.3.

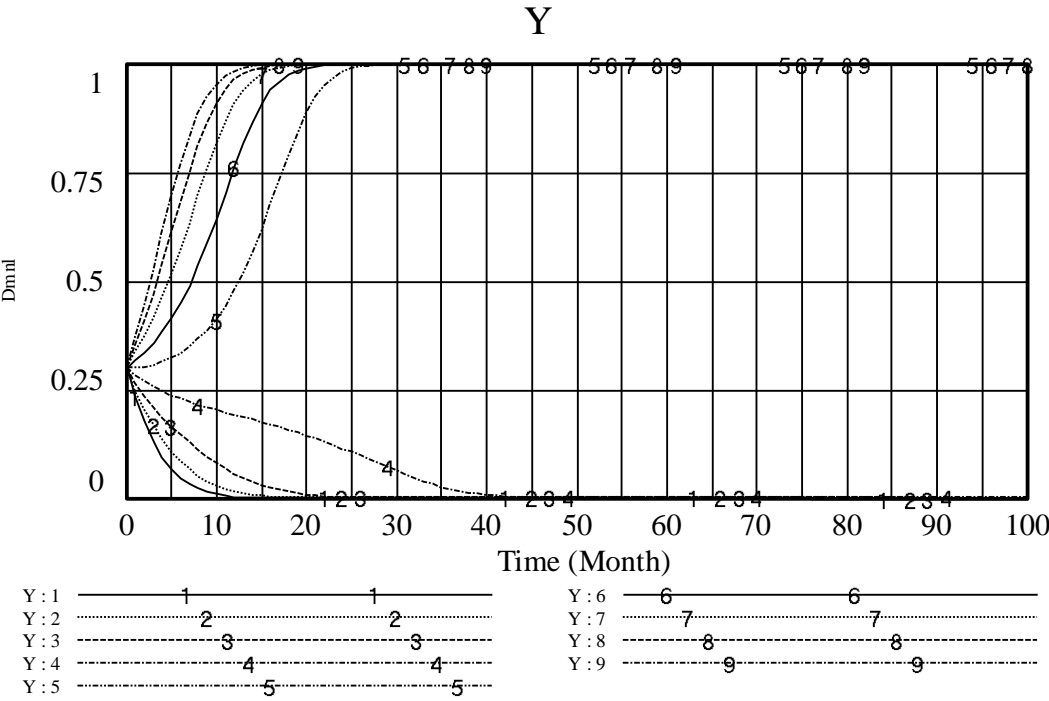

**Figure 4.** Simulation result of y when initial value of y is 0.3.

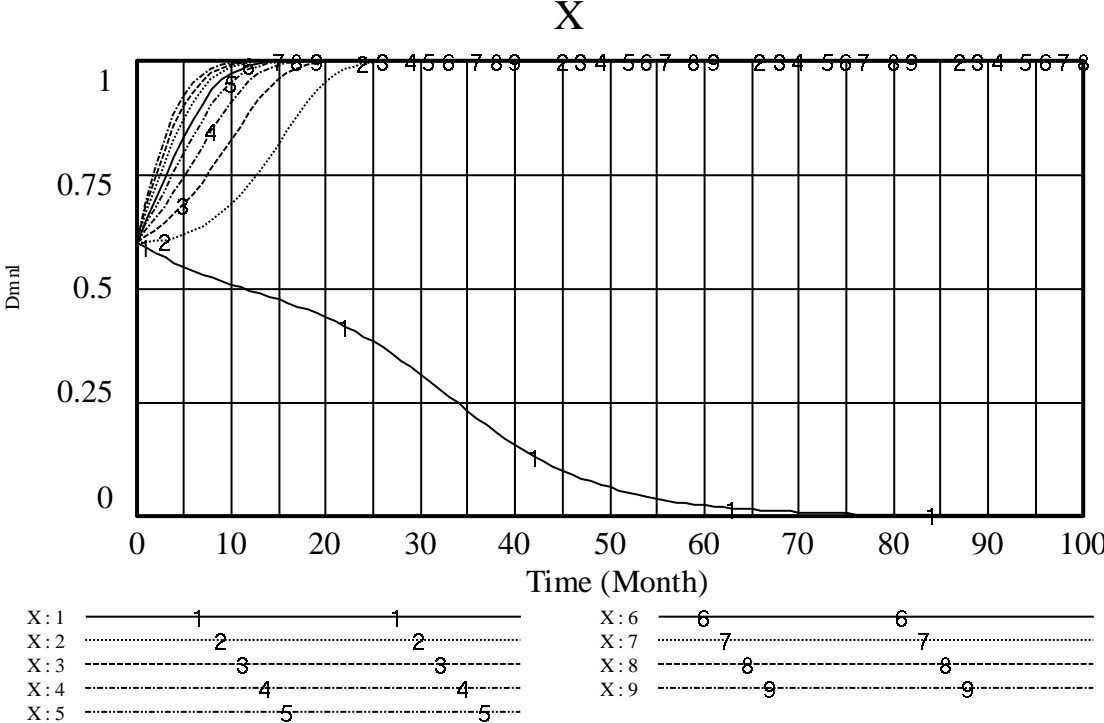

**Figure 5.** Simulation result of x when initial value of x is 0.6.

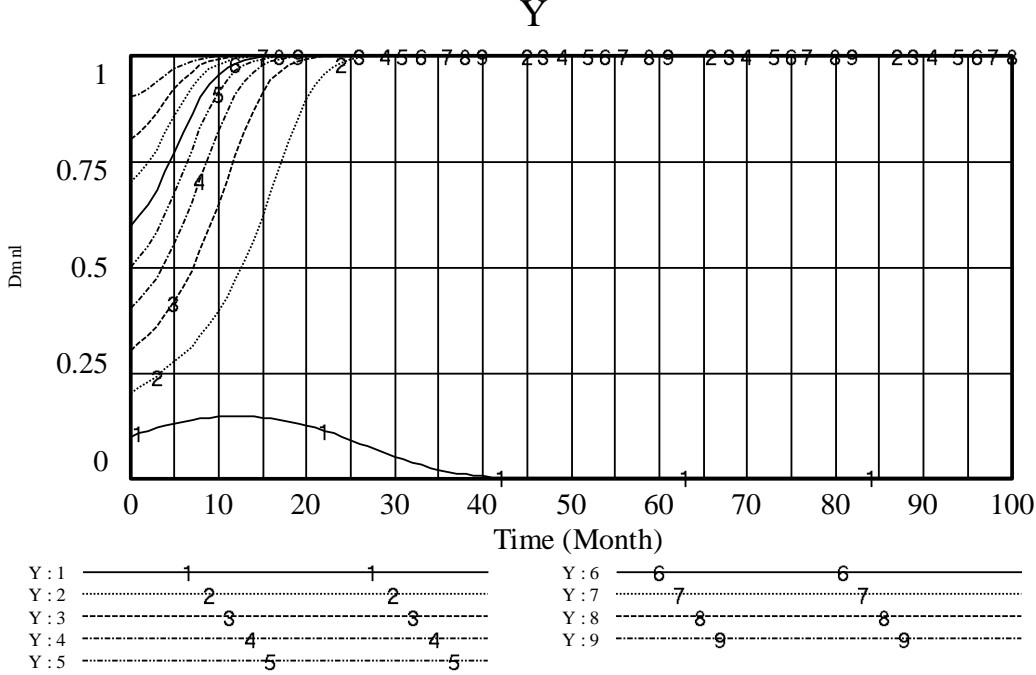

**Figure 6.** Simulation result of y when initial value of x is 0.6.

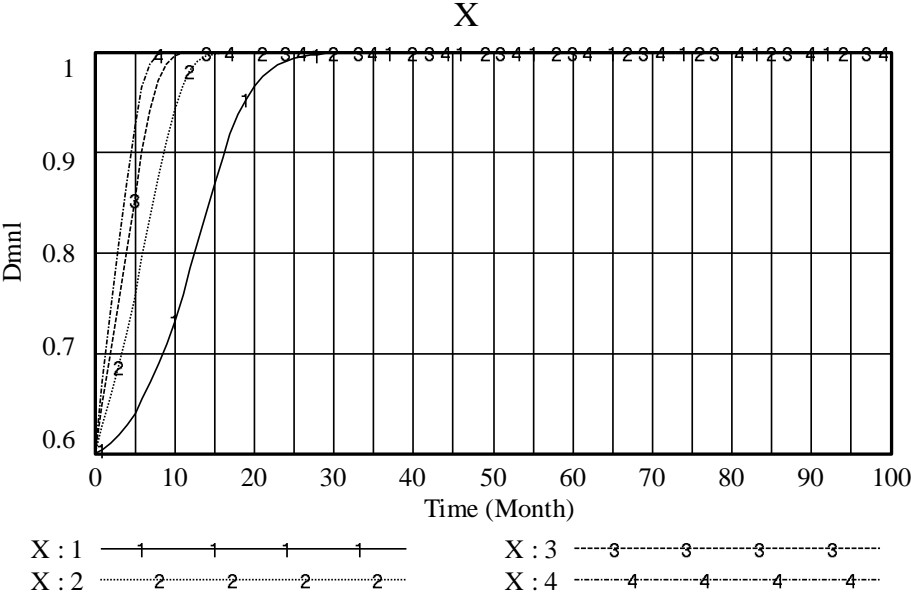

**Figure 7.** Simulation result of x when the amount of penalty, L changes.

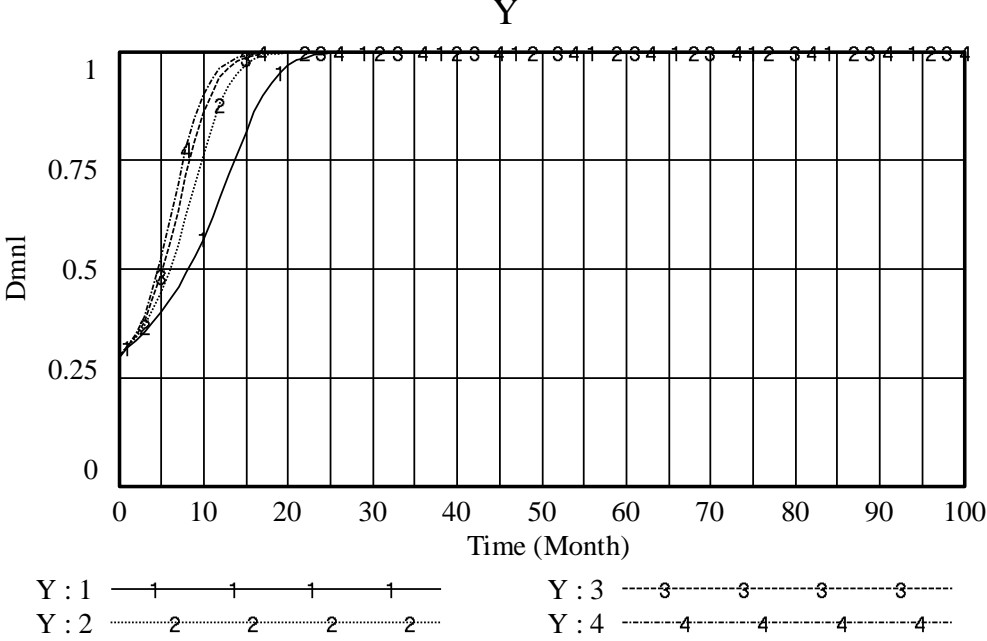

**Figure 8.** Simulation result of y when the amount of penalty, L changes.

## 5. Conclusions

We identified the problem of excessive growth and high risk of operation in China's Internet finance, and demonstrated the feasibility of supervising Internet financial risks by using the theory of the audit "immune system". Under the hypothesis of bounded rationality, we developed an evolutionary game model and computing experiment approach, and simulated the initial probability of different strategies adopted by both parties of the game. We also examined the influence of changing the penalty intensity of Internet financial institutions' violation on the outcome of the evolutionary game, and verified the effectiveness of audit supervision measures which were unable to be experimented upon in real Internet finance government audit supervision. Finally, we demonstrated how a change of strategies could bring about the evolution of penalty intensity and affect the gains and losses of the game parties, which were incalculable in real Internet finance government audit supervision. Based on

our simulation analysis results, we made policy suggestions on strengthening audit supervision and promoting its sustainable development from three aspects: strengthening the construction of Internet financial credit information system, improving Internet financial laws and regulations, and improving the early warning level of Internet financial credit risk.

First, we suggest that policymakers strengthen the construction of the Internet financial credit information system and bring the function of audit "immune system" in discovering problems into full play. According to Figure 2 and our evolution analysis of game model parameters, the behaviors of both parties in the game coordinate with each other. Internet finance complements the traditional financial system, and the credit information system is the cornerstone of the healthy development of Internet finance. At this stage, the Internet financial institutions, e-commerce platforms and the credit information agencies do not exchange information. The result is that the coverage of the current credit information agency is not extensive enough and cannot meet the needs of Internet finance. Thus, there is an urgent need for both the creditors and borrowers on the Internet finance platforms to establish an online credit information system. The construction of an Internet financial credit information system can help prevent Internet financial operation risks and guarantee compliance. At the same time, with the perfection of an online credit information system, the capability of discovering problems on the part of financial audit "immune system" can be enhanced and the efficiency of financial audit supervisory departments can be greatly improved.

Second, we suggest that policymakers improve the Internet financial laws and regulations, and bring the function of defense of the audit "immune system" into full play. We have demonstrated that changes in the penalty amount influence the decisions made by both Internet financial institutions and financial audit supervisory departments. Thus, the Internet financial supervision agents should strengthen the restrictions imposed by laws and regulations. Internet finance is a brand new industry, the legal relationships of which are much more complex than those of the traditional financial industry. The legal definitions of related parties are yet to be improved. As a result, the fundamental laws and regulations regarding Internet finance must be improved to provide guidance for solving basic problems. Strengthening the self-regulation and supervision of the industry will also help. For Internet finance, which relies greatly on financial information, industry associations act as an important bond between government supervision and platform operations. Rules and regulations made by industry associations can lead member entities to voluntarily comply with regulations, and promote the healthy interaction and development of the financial industry and the local economy.

Finally, we suggest that policymakers improve the early warning level of Internet financial credit risk to prevent risks though the audit "immune system". Our evaluation of Internet finance government audit supervision has shown that it is impossible for both parties of the game to make one-time decisions that maximize their own interests. As shown in Figures 3 and 4, their decisions change over time; audit (the most effective way of supervision in economic activities) plays a crucial role in preventing Internet financial risks and ensuring compliance. A scientific early-warning system could clarify the focus of audit, reduce the aimlessness of audit, and improve audit efficiency. On the one hand, integrating audit with Internet financial supervision helps effective disclosure of information and the improvement of Internet financial risks' early warning level. On the other hand, audit can help improve the risk-management system and the safety of funds, promote the self-discipline of Internet financial platforms, and promote the healthy development of the financial industry. A higher level of early warning can shift the focus of risk management and pay attention to Internet financial risks before and during transactions, and thus minimize losses.

**Author Contributions:** Conceptualization, S.G. and H.L.; methodology, H.L.; formal analysis, S.G.; writing—original draft preparation, H.L.; writing—review and editing, S.G. All authors have read and agreed to the published version of the manuscript.

**Funding:** This research receives funding from the School of Government Audit, Nanjing Audit University.

**Conflicts of Interest:** The authors declare no conflict of interest.

## Appendix A

The evolutionary game matrix in this paper contains 7 variables:

1.  $I_1$: the income from compliance operation of Internet financial institutions;
2.  $C_1$: expenditure cost of compliance operation of Internet financial institutions;
3.  $D_1$: extra rewards given by Internet financial institutions under the condition of compliance with business operations;
4.  L: punishment for non-compliant operation of Internet financial institutions;
5.  The above variables mainly describe the operation of Internet financial institutions, and relevant data can be referred to the P2P industry data provided by the "home of online loans".
6.  $C_2$: expenditure cost of financial audit supervision by financial audit supervision department for Internet financial institutions;
7.  $I_2$: social and economic benefits generated by mutual cooperation between financial audit supervision departments and Internet financial institutions in audit supervision;
8.  $D_2$: networked financial institutions maintain a state of compliance operation, which will create a good ecological environment for financial operations and reduce the working pressure of audit supervision for the financial audit department, thus bringing opportunities and benefits to the department.

The above variables are mainly related to the supervision of the audit department, and relevant data can be referred to the statistics in the special governance process of Internet finance by local audit institutions.

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
