# Peer review of "Research on Audit Supervision of Internet Finance"

_ijfs, doi:10.3390/ijfs8010002_

Round 1
Reviewer 1 Report
Although the topic is targeted and specific to the Chinese market, the bibliography can be improved especially on the broader references to different theories and research instruments. For example, as a market failure is clearly identified, the authors could elaborate more on the literature on information failure. This can be valid for all the references made to literature in general, more works should be cited so that the reader, should he/she needed more detailed information, could refer to the source.
Author Response
Reviewer #1: Great thanks to review’s comments which we definitely agree. We have improved the bibliography on a broader references to different theories and research instruments by adding the bibliography about the problems and risks of internet finance. Reference[4][5][6][7] is added to illustrate scholars' research on the asymmetric risk of Internet financial information when a market failure is clearly identified.
Reviewer 2 Report
There is quite big research done by this paper with certain proposed suggestions. The paper is highly recommended for expert readers, so the impact of the paper will be not so wide, but still very interesting results.
Author Response
Reviewer #2: Great thanks to review’s kind comments. It is true that the impact of the paper will not be so wide because this paper focus on audit supervision of internet finance. It is also worthwhile pointing out that with the rapid development of Internet financial in China, the risk supervision of online loan platform is increasingly attracting the attention of the state and the people. We hope our research on audit supervision of internet finance in China will provide useful perspective for studies on worldwide finance supervision.
Reviewer 3 Report
Refer to the attachment

Author Response
Reviewer #3: Great thanks to review’s comments which we definitely consider will help us improve the quality of our paper. For the literature review related comments, we have added literatures with data about these variables and improved the literature review part. Researches in this field in China and other countries are summarized and commented respectively. Two literatures, Mr Stiglitz and Weiss, 2016 and Yum, Lee, 2015, are added to illustrate the theoretical and practical contributions of other countries in the field of P2P risk and regulatory research in recent years.
Also, we have edited and revised accordingly the manuscript with the help of a professional English-native writer and most nonstandard expressions are modified and improved.
Currently, seven main variables are used in establishing game model with an evolutionary game theory, we illustrated them in the paper together before evolutionary game matrix instead of using an appendix.